# RE-BENCHMARKING OUT-OF-DISTRIBUTION DETECTION IN DEEP NEURAL NETWORKS

## ABSTRACT

Out-of-distribution (OOD) detection is a key challenge for making machine learning models robust in the real world, where we want models to be aware of uncertainty outside their training data distribution. Despite the rapid development of existing OOD detection algorithms, their experimental settings are usually inconsistent, e.g., datasets, evaluation metrics, model selection, implementation choices. In this paper, we aim to understand OOD detection fundamentally and provide a comprehensive benchmarking of the current state of the art OOD detection methods in a consistent and realistic evaluation setting. This benchmarking contains a serious of datasets split, model selection criteria and OOD detection algorithms.This experimental framework can be easily extended to new algorithms, datasets, and model selection criteria. We conduct extensive experiments on this benchmark and re-evaluate algorithms contained in this benchmark.

## 1 INTRODUCTION

Deep learning has achieved superior performance when the training and test data are sampled from the same distribution (Krizhevsky et al., 2012; He et al., 2016; Redmon et al., 2016), i.e., independent and identically distributed (i.i.d.). However, the real world is open and presents diverse inputs for modern deep learning models. Out-of-distribution (OOD) detection is of great importance to AI safety, which aims to determine whether an input is different from its training data and reject them (or safe handling by human users) (Yang et al., 2021; Hendrycks & Gimpel, 2017).

Extensive recent research endeavors are devoted to the problem of out-of-distribution detection (Liang et al., 2018; Hsu et al., 2020; Huang et al., 2021; Ming et al., 2022; Sun et al., 2022; Meinke & Hein, 2020), including softmax-based score and generative-based methods. The goal of out-of-distribution detection is to effectively separate between ID and OOD data. Most of the previous works with discriminative models focus on deriving score criterion of the probability density of OOD samples from different space of modern deep neural networks, e.g., output space (Hendrycks & Gimpel, 2017; Liu et al., 2020), feature space (Lee et al., 2018) and gradient space (Huang et al., 2021). Generative-based methods leverage deep generative models for OOD detection and assign a high likelihood to test OOD data (Ren et al., 2019; Schirrmeister et al., 2020; Xiao et al., 2020).

Although a plethora of OOD detection algorithms has been proposed recently, they are evaluated based on different OOD datasets, evaluation metrics, and model selection criteria. Inspired by standard benchmarks such as ImageNet (Russakovsky et al., 2015), DomainBed (Gulrajani & Lopez-Paz, 2021), we propose a comprehensive testbed for out-of-distribution detection. This benchmark includes 5 OOD datasets, 5 baseline algorithms, and three evaluation criteria. In this work, we provide a framework to reproduce experimentation in OOD detection and propose a simple yet effective method for adaptive model selection in OOD detection. Then, we carefully implement five OOD detection algorithms on five representative OOD datasets and three evaluation criteria and we reveal that the changeable threshold to separate IID and OOD data may be a possible reason to sub-optimal OOD detection performance, while existing works lacks the in-depth analysis on model selection and evaluation criteria of OOD detection algorithms. Based on the experiments and analysis in this work, leading to the recommendation for OOD research: *The threshold of a OOD detection algorithm should be stable against model selection criteria.*

Based on the observations and analysis, we make several recommendations for future OOD detection research:

- We propose a framework comprehensively benchmarking OOD detection algorithms in a rigorous and realistic evaluation setting.
- We point out the weakness of previous benchmark methods and re-evaluate algorithm performances under different data-splitting methods.
- We observe the unfairness of existing metric in OOD detection.
- We understand OOD detection fundamentally and illustrate the importance of threshold that separating IID and OOD data for model selection in finding optimal OOD detection model.

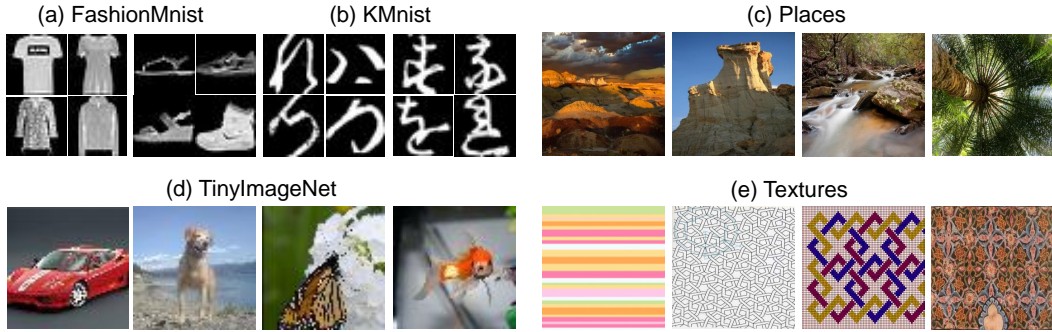

Figure 1: Visualization of typical examples of widely used OOD detection datasets.

## 2 RELATED WORKS

### 2.1 OOD DETECTION ALGORITHMS

GradNorm (Huang et al., 2021) employs the vector norm of gradients, which backpropagated from the KL divergence between the softmax output and a uniform probability distribution, to derive OOD scores for detecting OOD data. MOS (Huang & Li, 2021) proposes a group-based OOD detection framework with a scoring function MOS to decompose the large semantic space into smaller groups with similar concepts, which allows simplifying the in distribution and OOD data decision boundaries and thus enhancing OOD detection especially in a large-scale setting. MLP (Hendrycks & Gimpel, 2017) presents a simple baseline that utilizes probabilities from softmax distributions for detecting OOD data. Generalized ODIN (Hsu et al., 2020) proposes two strategies, the decomposed confidence and the modified input preprocessing for improving OOD detection, and does not need OOD data for tuning the model.

ODIN (Liang et al., 2018) separates the softmax score distributions and achieves more effective OOD detection by using temperature scaling and adding small perturbations to the input. The work (Liu et al., 2020) uses an energy score to distinguish ID and OOD data, that is theoretically aligned with the probability density of the inputs. Energy can be used as both a scoring function of any pre-trained classifier and a trainable cost function to fine-tune the model. Outlier Exposure (OE) (Hendrycks et al., 2019) utilizes auxiliary data to help OOD detectors work well on unseen distributions of OOD data. Mahalanobis (Lee et al., 2018) proposes a confidence score based on the Mahalanobis distance to obtain the class conditional Gaussian distributions for detecting out-of-distribution. Likelihood Ratio (Ren et al., 2019) proposes a deep generative model based approach and a likelihood ratio method for OOD detection.

**OOD detection using auxiliary outlier data** assume that auxiliary outlier data is completely separable from ID data. ViM (Wang et al., 2022) present a virtual-logit matching OOD score method that combines feature space class-agnostic score and ID class-depedent logits. DUQ (Van Amersfoort et al., 2020) is based on RBF networks to estimate uncertainty in a single forward pass. WOODS (Katz-Samuels et al., 2022) leverages wild mixture data and utilizes a machine learning classifier in natural habitats. Metric Learning (Masana et al., 2018) makes use of metric learning

instead of cross-entropy loss to divide prediction on the learned classes. Generative OOD Samples (Vernekar et al., 2019) proposes an algorithm to generative effective OOD samples based on a manifold learning network. GOOD (Bitterwolf et al., 2020) focus on certificating worst case guarantees by applying low confidence at OOD point for OOD detection.

## 2.2 BENCHMARKS

There are several public available datasets for benchmarking OOD detection algorithms: FasionM-NIST (Xiao et al., 2017), Kuzushiji-MNIST (Clanuwat et al., 2018), Texture Dataset (Cimpoi et al., 2014), Places (Zhou et al., 2017), CIFAR-10, CIFAR-100 (Krizhevsky et al., 2009), SVHN (Netzer et al., 2011), LSUN (Yu et al., 2015), ImageNet-O (Hendrycks et al., 2021; Srivastava et al., 2022). In this work, we use the standard split for each ID and OOD datasets. There is a widely used common benchmark in recent years: CIFAR-10, CIFAR-100 as ID data, SVHN, LSUN, Places365, Textures as OOD data. (standard ResNet-20 (He et al., 2016) network). CAOS Benchmark (Hendrycks et al., 2022) and MOS (Huang & Li, 2021) (Google BiT-S Kolesnikov et al. (2020) models) utilizes a large-scale that leverage ImageNet-21K to evaluate multilabel anomaly detectors: ImageNet as ID dataset and iNaturalist, SUN, Places, Textures as OOD datasets. These are commonly used evaluation metrics in existing OOD detection benchmarks: AUROC (Hendrycks & Gimpel, 2017), AUPR (Hendrycks & Gimpel, 2017), False-positive rate (FPR95), and Detection Accuracy.

Recently, a new codebase called OpenOOD is proposed (Yang et al., 2022), in which nine benchmarks from neighboring fields such as anomaly detection and open set recognition are considered. Yang utilizes unified hyperparameters and architecture choices aiming to ensure fairness among different methods and compares the performances of methods under various dataset benchmarks. However, no one focuses on multiple datasets nor spliting datasets.

## 3 NEW BENCHMARK

Here we make a detailed introduction of datasets, baseline algorithms, and the methods of split of datasets used in our new benchmark. These datasets and algorithms are selected according to prior works with some slight changes.

## 3.1 OOD DETECTION

Threshold appears crucial in OOD detection since it is used to discriminate ID data and OOD data. It serves as the boundary between ID data and OOD data. If the threshold is too small, then the model will mistake OOD data as ID data and opposite if the threshold is too big. Thus, once other factors such as datasets, algorithms, model selection, and dataset split are determined, the threshold becomes the most important element that affects OOD detection. The task of improving the performance of OOD detection turns into the task of finding the optimal threshold. In the setting of optimal OOD detection, the threshold of a OOD detection algorithm should be stable against model selection criteria.

1) The distance between threshold of a OOD detection algorithm and the expectation of optimal threshold is minimized. The less the distance is, the better performance will the model show. However, it is hard to get the optimal threshold.

2) The variance of threshold is minimized. In real world, there often exist more than one datasets. Hence, one stable threshold can be used for multiple datasets, which can save much time without sacrificing the performance of OOD detection. In our benchmark, we consider the threshold as well and compare different thresholds in all cases.

## 3.2 DATASETS

Previous works use a certain dataset as ID data, e.g., CIFAR-10 (Krizhevsky et al., 2009), CIFAR-100 (Krizhevsky et al., 2009), and ImageNet-21K Hendrycks et al. (2022) and some other datasets are then fixed as OOD data. In our benchmark, we choose five commonly used datasets. However, these datasets are used as both ID data and OOD data. In different groups, the same dataset may play different roles. These datasets do not overlap with each other. TinyImageNet (Le & Yang, 2015)

contains 200 categories of different creatures with 500 training images in each category. We use the entire training data in our experiment. FashionMNIST (Xiao et al., 2017) is a dataset about fashion clothes, including 60,000 training images within 10 classes. We pick out the training data from each class in order to form the dataset. KMNIST, whose full name is Kuzushiji-MNIST (Clanuwat et al., 2018), is similar to FashionMNIST (Xiao et al., 2017) since they share the same number of images and categories. The difference is that KMNIST focuses on Japanese characters. According to the similarity of KMNIST and FashionMNIST, we choose images from KMNIST in the same way as we choose images from FashionMNIST. Places365 consists of 180,000 images from 365 classifications of scene. We use the subset of 10,000 images among 50 manually selected classes. Textures (Cimpoi et al., 2014) is a textural dataset containing 5,640 images under 47 categories collected from real world. We put all these data from Textures into our experiments.

## 3.3 BASELINE ALGORITHMS

We choose five baseline algorithms of OOD detection, MSP (Hendrycks & Gimpel, 2017), ODIN (Liang et al., 2018), OE (Hendrycks et al., 2019), Energy (Liu et al., 2020), and Grad-Norm (Huang et al., 2021), in our benchmark. Their performances and effectiveness have been proved in former works. However, one baseline, GradNorm, requires some changes. This baseline calculates scores from parameters of the layer of the model. Accordingly, those datasets which have more classes will have higher grad_norm scores even if they are OOD datasets, which contradicts our hypothesis that ID datasets have higher scores than OOD datasets. After we get the grad_norm scores, we divide these scores by the classes of their datasets. Thus, we change grad_norm into mean grad_norm and eliminate the impact of data types.

## 3.4 METRICS

As for the metrics, we not only use the traditional criteria such as FPR95, AUROC (Hendrycks & Gimpel, 2017) and AUPR (Hendrycks & Gimpel, 2017), but also choose F1_score, accuracy, precision rate as well as recall rate. The accuracy shows if the model can judge whether the data belongs to ID dataset or OOD dataset correctly. The precision rate is the rate represents the proportion of OOD data correctly judged in all data which is considered as OOD data. The recall rate represents the proportion of OOD data which is correctly judged in all OOD data. FPR95 represents the probability that OOD data is mistaken as ID data when the accuracy of classifying OOD data reaches 95%. AUROC and AUPR are two basic metrics in classification problems. These two metrics show the capacity of the model to distinguish different classifications. In our benchmark, we use F1_score, FPR95, AUROC as well as AUPR as the metrics to evaluate the performance of OOD detection. Among these metrics, F1_score, AUROC and AUPR are higher when models discriminate OOD data from ID data better, while FPR95 is in reverse proportion to the performances of models in OOD detection.

## 3.5 DATASETS SPLIT

Prior works have focused on deriving score criterion of data from different aspects until Gulrajani and Lopez-Paz recommended a new benchmark in which model selection is taken into account in OOD detection. However, datasets split is an importance factor in OOD detection as well. Thus, in spite of the traditional method of datasets split, we propose another two methods of datasets split.

**Case1: Standard method**

The existing method of OOD detection is to select one dataset as ID dataset and choose another dataset as OOD dataset. We use the same method in case 1. Firstly, we set five clusters. Each cluster contains one dataset, which is used as ID dataset. These clusters are distinguished according to their ID dataset. For example, the ID dataset in the first cluster is FashionMnist while that in the second cluster is KMnist. Hence, there remain four datasets in each cluster. Ulteriorly, we divide each cluster into four groups. Each group includes one dataset selected among those four remaining datasets. For example, the four groups in the first cluster consist of KMnist, Places, Textures and TinyImageNet respectively. Then we get twenty groups of diverse ID data and OOD data. After dividing these groups, we train models for each cluster using its ID dataset. When we get these five models, we can start our work of OOD detection. Since each cluster has one model and four groups,

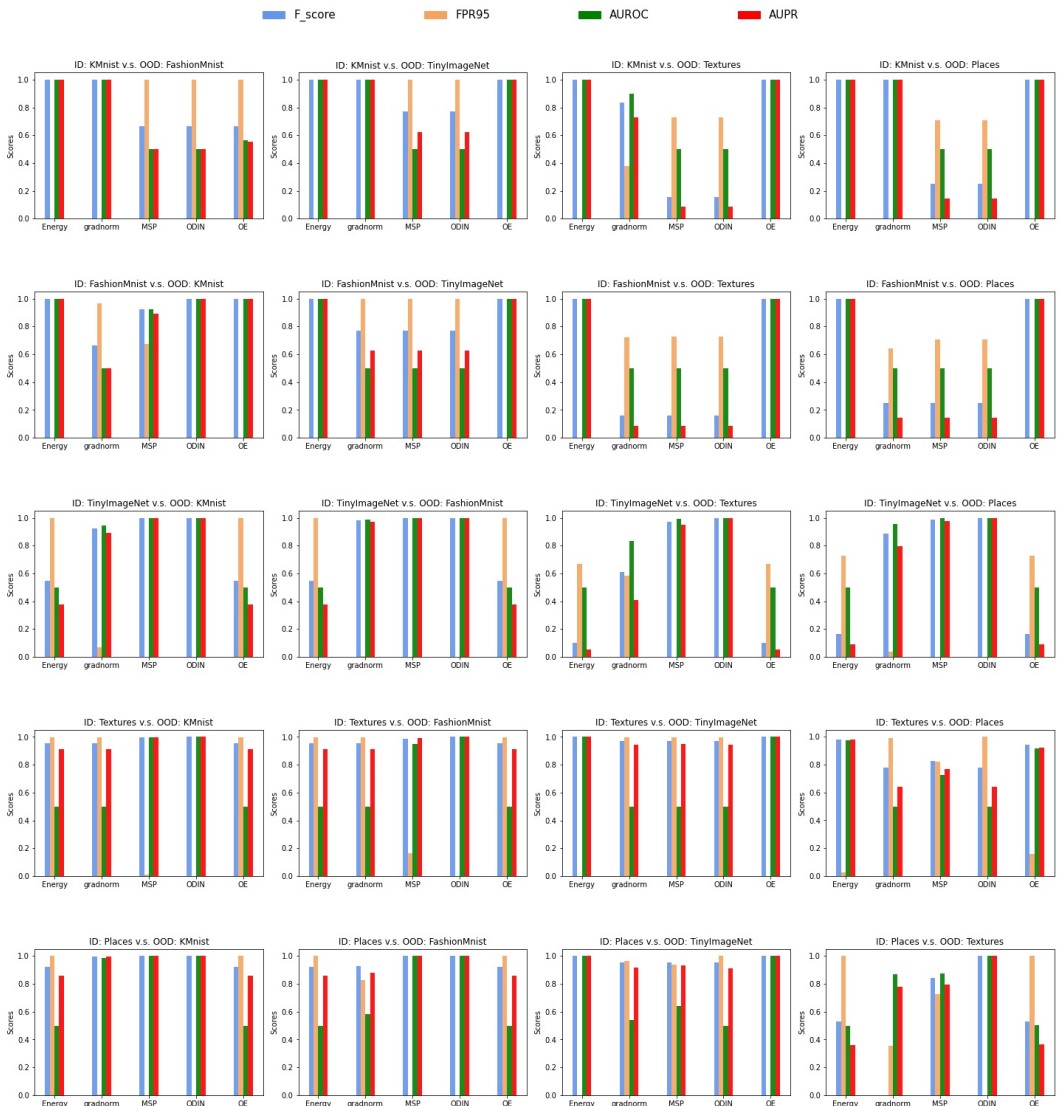

Figure 2: Comparisons of state-of-the-art OOD detection algorithms in terms of different evaluation metrics of Case 1. We plot results of the three metrics and the thresholds in case 1. The five baselines in the same group are put into the same figure.

we use this model to detect OOD data for these four groups. For each group, we calculate scores of all methods, such as grad_norm and energy, for each batch of data in both ID dataset and OOD dataset. After we get the scores of one group, we use these scores to discriminate ID data and OOD data in this group, and then choose the score which has the best performance in OOD detection as the result for this group. At the end, we obtain twenty groups of results for each method.

**Case2: Multiple ID datasets and multiple OOD datasets**

In this case, we choose multiple datasets as ID datasets and multiple datasets as OOD datasets in a group. For each group, we select three of the five datasets as ID datasets and use the other two datasets as OOD datasets. Different groups contain diverse combinations of datasets. In each group, we utilize ID datasets in order to train a model which belongs to this group. Nevertheless, these three ID datasets are used one by one. By contrast, the model is trained by a batch of data from one dataset at one time. Then, another batch of data from another ID dataset is sent into the model. When we get the models in all groups, we start the work of OOD detection. We calculate the scores of all methods in each group, and then use them as the threshold to detect OOD data in their groups.

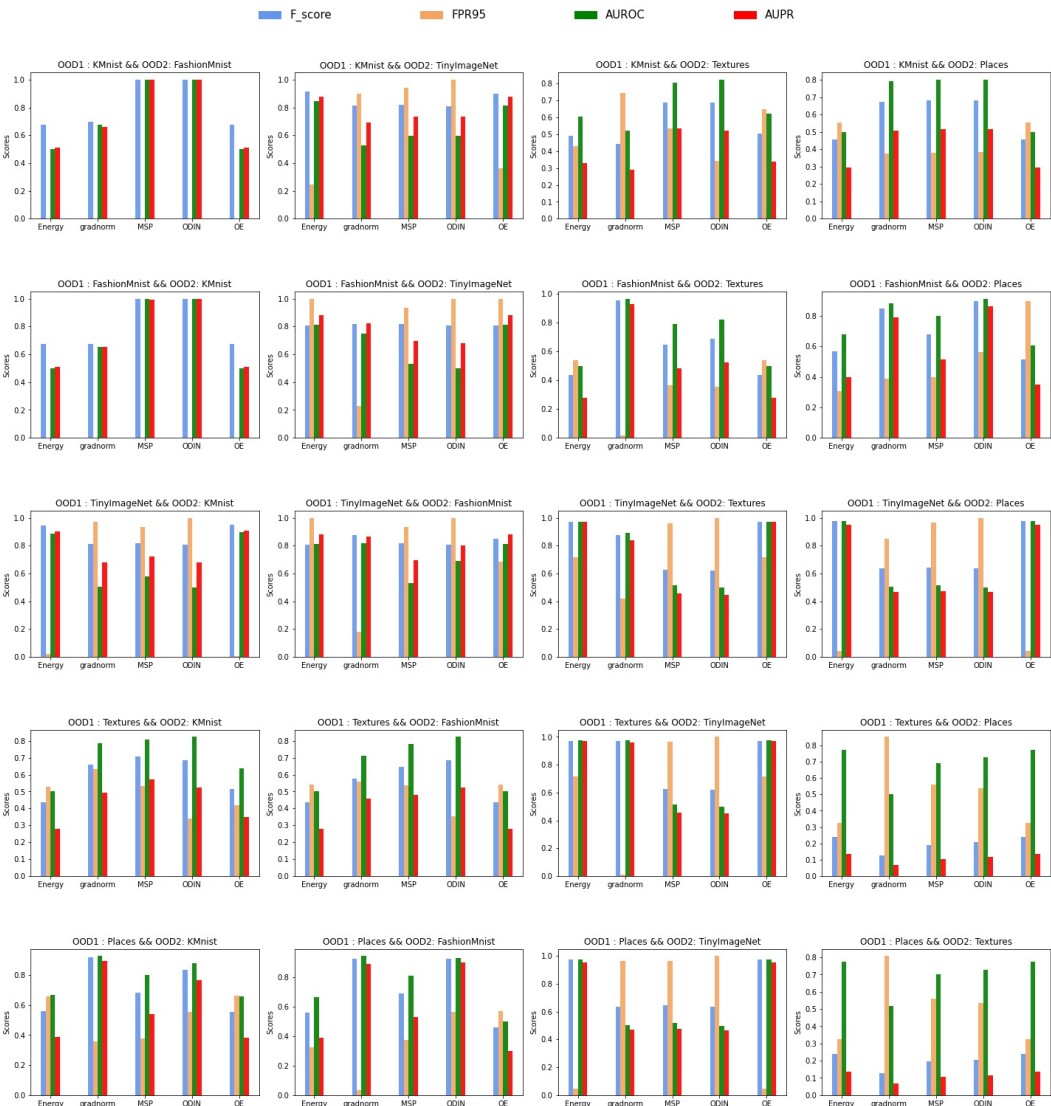

Figure 3: Comparisons of state-of-the-art OOD detection algorithms in terms of different evaluation metrics of Case 2.

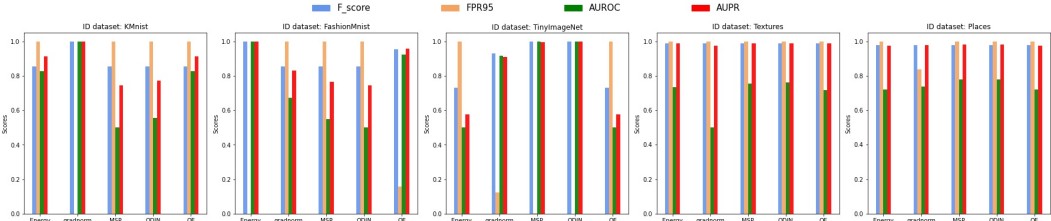

Figure 4: Comparisons of state-of-the-art OOD detection algorithms in terms of different evaluation metrics of Case 3.

For each group, we compare the performance of each threshold and select the optimum threshold which performs the best in OOD detection.

**Case3: All datasets in a group**

Here in this case, we combine the selection of datasets of the first two cases. We use one dataset as ID dataset and multiple datasets as OOD datasets in a group. At the beginning, we select one dataset among these five datasets for each group. These groups are different from each other since each group contains a different dataset, which is used as ID dataset in this group. Then for each group, we make the remaining four datasets OOD datasets. When we finish grouping these datasets, we use the ID dataset in each group to train a model for the group which the dataset belongs to. After models are trained, we calculate the scores of all methods for each group using its ID dataset and OOD datasets. However, unlike what we do in the first case, we make the scores of different groups in common use among all groups. The scores from these five datasets compose the threshold set. Then, we test all scores in the threshold set and choose the one that shows the optimum performance for each group.

Table 1: The average results of the same method under the same metric in the second case

| Algorithms \ Metrics | F1_score | AUROC | FPR95 | AUPR |
|---|---|---|---|---|
| Energy | 0.83 | 0.77 | 0.42 | 0.79 |
| GradNorm | 0.82 | 0.73 | 0.53 | 0.76 |
| MSP | 0.74 | 0.76 | 0.51 | 0.72 |
| ODIN | 0.74 | 0.75 | 0.44 | 0.74 |
| OE | 0.79 | 0.75 | 0.48 | 0.76 |

Table 2: The average results of the same method under the same metric in the second case

| Algorithms \ Metrics | F1_score | AUROC | FPR95 | AUPR |
|---|---|---|---|---|
| Energy | 0.65 | 0.72 | 0.42 | 0.57 |
| GradNorm | 0.70 | 0.72 | 0.47 | 0.62 |
| MSP | 0.67 | 0.70 | 0.61 | 0.55 |
| ODIN | 0.70 | 0.73 | 0.63 | 0.60 |
| OE | 0.65 | 0.72 | 0.45 | 0.56 |

Table 3: The average results of the same method under the same metric in the second case

| Algorithms \ Metrics | F1_score | AUROC | FPR95 | AUPR |
|---|---|---|---|---|
| Energy | 0.91 | 0.76 | 0.80 | 0.89 |
| GradNorm | 0.95 | 0.77 | 0.59 | 0.94 |
| MSP | 0.93 | 0.72 | 0.80 | 0.90 |
| ODIN | 0.88 | 0.72 | 0.80 | 0.90 |
| OE | 0.90 | 0.74 | 0.83 | 0.88 |

# 4 EXPERIMENTS

## 4.1 PREPARATION BEFORE EXPERIMENTS

We use ResNet18 which has been pre-trained as the model in all our experiments. We separate all images based on the classes and give each image a corresponding label because what we need is supervised learning. Before our experiments start, we resize all images to 224 × 224. For each dataset to be trained in the models, it is split into training set and validation set. Training set accounts

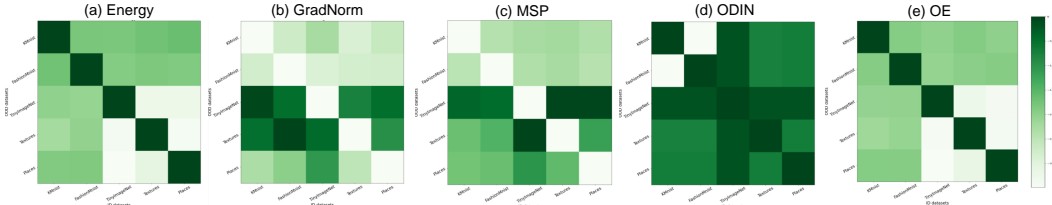

Figure 5: F1_score thresholds of different metrics in case 1. Each figure represents one metric in case 1. The figure is divided into several small areas, each of which shows the threshold of a group in case 1. The diagonal areas means ID dataset is equivalent to OOD dataset. We stipulate thresholds in diagonal areas are 0 since they are not used in our experiments. (Better viewed in the zoom-in mode)

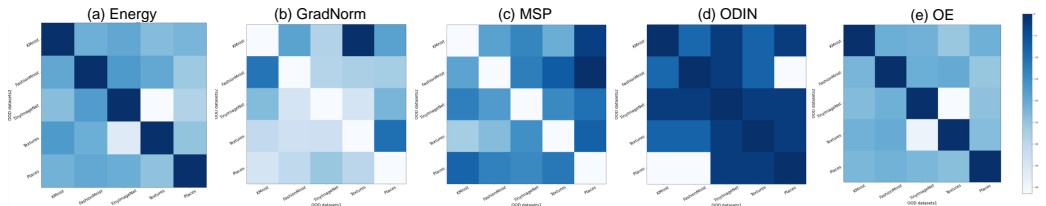

Figure 6: F1_score thresholds of different metrics in case 2. We use the same method as the one in the first method to plot thresholds in case 2. (Better viewed in the zoom-in mode)

for 80% of the data while the remaining 20% is used for validation. The images are sent into models as a batch at one time and the batch size is set to 8.

## 4.2 EXPERIMENT RESULTS AND ANALYSIS

The main results of our experiments are shown in the figures above. We print the highest F1_score, the highest AUROC, the highest AUPR and the lowest FPR95 of different groups of each baseline algorithm using these three methods. For each method, we calculate the average scores of all groups in the same algorithm under the same metric, which are used to appraise the performance of OOD detection with the corresponding algorithms. We then make a comparison of the scores among algorithms. Afterwards, we attain the following results:

**Previous benchmark method exaggerates the effectiveness of OOD detection algorithms** Previous researchers use the same but simple method of splitting datasets, one as ID dataset and another as OOD dataset, to test their algorithms. However, according to the results that we get from these three methods, the traditional method cannot reflect the real effectiveness of algorithms. From Table 1, Table 2, and Table 3, the results vary widely among methods. For example, the result under AUPR in Energy using the traditional method is 78.88% while that result reduces to 56.54% using the second method to split datasets and increases to 89.04% in the last method. Even though we get high AUPR scores in the third method, the FPR95 scores increase relatively. The statistics above show that algorithms perform worse when multiple datasets form ID datasets and OOD datasets, which represents the effectiveness of algorithms. Since traditional method only focuses on single ID dataset with single OOD dataset, the real effectiveness of these algorithms is overstated.

**Among all methods, GradNorm outperforms other algorithms on various benchmark settings but is still not robust** Figure 1 shows the performance of different algorithms in the first method. AS we can see, Energy seems the best algorithm since it has the optimum results under per metric. However, when it comes to the other two methods, to our surprise, GradNorm becomes the best algorithm. In the second method, GradNorm reaches 62.496% under AUPR while in the last method, it performs the best under all metrics. Thus, GradNorm has more potential when OOD detection includes more than one dataset. It seems the more data included in OOD datasets, the better GradNorm performs compared to other algorithms. We assume two reasons to explain its strength. The first one is that the gap of gradients between ID data and OOD data can be enlarged with the addi-

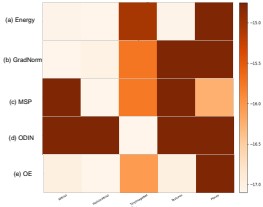

Figure 7: F1_score thresholds of different metrics in case 3. Unlike the two cases above, we only have five groups distinguished by ID dataset. We use ID datasets as the x-axis and plot thresholds of each group in different metric figures. (Better viewed in the zoom-in mode)

tion of datasets, which enables models to discriminate ID data and OOD data more easily. Besides, considering models in case 1 are shared in case 3, they should have similar performances. However, GradNorm seems better in case 3. Then we propose the second assumption that GradNorm may be adept in detecting OOD data and perform worse in classifying ID data. Though GradNorm shows better performance than other algorithms, it still has some weakness. As we can see in Figure 5, Figure 6, and Figure 7, the thresholds got in GradNorm are not stable, which means GradNorm is not robust and may have poor performance when used in different groups of datasets.

**AUPR might not be a good indicator for OOD detection** Former researchers use AUROC, FPR95, and AUPR as basic metrics to evaluate their algorithms. In our experiment, we use these metrics as well. Since the calculation of AUROC as well as AUPR has no relation with thresholds, they only reflect the effectiveness of the classifier. However, after comparison of results of three methods, we find that AUPR scores vary widely when tested in different methods while AUROC scores show few differences, which means the effectiveness of the same classifier is different under these two metrics. We suppose that this phenomenon is resulted by the unbalanced datasets. AUPR represents area under PR curve, which has a great change when data is unbalanced, the propotation of positive data or negative data is much larger than the other one. Thus, the datasets selection and split method will affect AUPR scores. In other words, AUPR scores cannot reflect the real performance of algorithms and seems unfair as a metric in OOD detection.

## 5 CONCLUSION

In this paper, we re-test five OOD detection algorithms in three dataset splitting methods including the traditional method and two newly proposed methods. We summarize the following insights: **First,** Former benchmark method of dataset splitting magnifies the real effectiveness of OOD detection algorithm. Algorithms show worse performances when the group contains multiple datasets. However, the traditional method forgets to take this circumstance into account. **Second,** GradNorm is the best algorithm among the five algorithms when there exist more than one dataset in a group. Even though the performance of GradNorm cannot match that of Energy in the traditional method, it transcends all other algorithms in the second and last method. In other words, the real effectiveness of GradNorm outperforms the other algorithms. **Last,** AUPR seems to be an unfair indicator in OOD detection though it is widely used. AUPR scores rely on the proportion of OOD data. Thus, AUPR may vary widely when using different groups of datasets, which is unfair to evaluate algorithms. We hope our work can remind others that there still remain details which can be optimized in OOD detection work. Even though some parts of OOD detection work have been widely utilized, they may be unjustified and can be improved.

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
