# OpenReview forum: "Re-Benchmarking Out-of-Distribution Detection in Deep Neural Networks"
_ICLR.cc/2023/Conference — Submitted to ICLR 2023_

### Official Review · Reviewer_UJzP · 2022-10-20

**Confidence:** 3
**Correctness:** 2
**Technical Novelty And Significance:** 1
**Empirical Novelty And Significance:** 1
**Recommendation:** 3

**Clarity, Quality, Novelty And Reproducibility:**

The paper is not clear, nor ready for submission. There are no experimental details so it cannot be reproduced either. The content is not novel: simple application of existing methods on several datasets without a thorough analysis afterwards.

**Strength And Weaknesses:**

Weaknesses:
1) The manuscript does not seem ready for submission at all. There are typos everywhere. For example, Tables 1, 2 and 3 have the exact same legend. I am not going to detail all of them but sentences like " Likelihood Ratio (Ren et al., 2019) proposes a deep generative" is another example of lack of proof-reading.
2) There are no experimental details so the work cannot be reproduced.
3) The content is not novel: simple application of existing methods on several datasets without a thorough analysis afterwards.

**Summary Of The Paper:**

This paper is on OOD detection and evaluate existing algorithms on several datasets. I am not elaborating as the manuscript does not seem ready for submission at all. Please see the weakness section.

**Summary Of The Review:**

I am voting for rejection as the manuscript does not seem ready for submission and the content is not novel and not reproducible.

---

### Official Review · Reviewer_ZnnK · 2022-10-23

**Confidence:** 5
**Correctness:** 1
**Technical Novelty And Significance:** 1
**Empirical Novelty And Significance:** 1
**Recommendation:** 1

**Clarity, Quality, Novelty And Reproducibility:**

In terms of clarity, the paper is very unclear, mainly on what is the contribution or difference with the state of the art. I think the authors meant their contribution to be the dataset splits, but from the paper I cannot understand what exactly happens in these splits, and without understanding that, I cannot validate the rest of the paper. Additionally I believe that the paper lies in faulty assumptions like that thresholds should be stable and unrelated to models or datasets.

In terms of originality and quality, Unfortunately  do not find original contributions in this paper.


**Strength And Weaknesses:**

Strengths
- Benchmarking OOD detection is a good idea overall, it is an often overlooked area of research, and the authors make a correct assessment that multiple datasets should be used. In this paper five datasets are used for evaluation.

Weaknesses
- The paper is difficult to read and ideas do not always follow clearly. Overall I had troubles understanding the writing of this paper, particularly  Section 3.5.
- The dataset split section (Sec 3.5) proposes three ways to split datasets, but after reading this section multiple times, I cannot understand exactly how this works. Case 1 is mostly clear, but Case 2 and 3 are not. In particular it is not clear to me how multiple datasets are used to train a single model, and if I am misunderstanding, how multiple models are used to perform OOD detection. The paper is very light on details, and I believe this would be better explained using mathematical formulations.
- The paper fails to make a good case for about its conclusions and contributions. Overall I do not find that the evaluation in this paper is correct and its conclusions are weak or irrelevant, particularly about the influence of thresholds or selection of methods to compare. There are too many question marks on the evaluation.
- One argument made in the paper is about thresholds, that "The threshold of a OOD detection algorithm should be stable against model selection criteria". Unfortunately I do not agree with the authors on this definition, the threshold for deciding ID or OOD usually depends on many factors, depending on the OOD detection methods, many methods use activations or other complex statistics, which do depend on the model architecture or dataset. Taking that the threshold should be stable as an assumption does not seem to be well justified, and the paper does not discuss further on this assumption.
- The paper does not make a clear definition of threshold (in Sec 3.1), the paper talks about threshold generically, and a mathematical model of the threshold is not provided, some simple definitions on how you use the threshold would suffice, to make it clear to the reader.
- The selection of OOD detection methods is not well justified. The authors do survey a good selection, but then select five methods without proper justification. In particular I think uncertainty-based methods should be there (like ensembles, dropout, DUQ, etc), and a justification on why those methods were chosen should be added.
- The selection of metrics is not well justified. The authors use F1 score, accuracy, precision, and recall, but they do not say why these metrics are necessary and what they are measuring in particular for the OOD detection problem.
- Also about metrics, AUPR and AUROC are threshold independent, since they use multiple thresholds, and other metrics like precision, recall, accuracy, and F1, do need a threshold, but the paper does not mention how the threshold was set to compute any of these metrics. The only sentence mentioning something related is "For each group, we compare the performance of each threshold and select the optimum threshold which performs the best in OOD detection" which is extremely vague.
- The paper mentions "previous benchmarks" but does not cite or refer to which benchmarks they are talking about. This severely undermines the connections of this paper to the state of the art and its scientific credibility.
- Considering all the doubts I have in the concept of the paper, its evaluation, I cannot consider the conclusions to be valid. Overall I do not find any relevant contributions to the state of the art.

Minor Issues
- Tables 1, 2, and 3 have exactly the same caption, it is not clear how these results are different. This is a major issue that affects how results can be interpreted.
- Paper does not have an appendix, since there are many details and explanations missing, some of them should go into the supplementary material. I note that the supplementary material exists, but it only proposes a new algorithm that is not evaluated in the paper, so functionally it is not an appendix to this paper.

**Summary Of The Paper:**

This paper is about benchmarking out of distribution detection algorithms, using multiple datasets.

The paper has no relevant contributions, due to lack of clarity in the paper (I cannot understand exactly what the authors propose as new benchmark or dataset split), and overall I do not believe the evaluation is valid, there are too many issues in this paper to speak of contributions.


**Summary Of The Review:**

I cannot recommend this paper for acceptance. There are many issues and weaknesses, the main body of the paper is not understandable, the selection of OOD detection methods and metrics is not appropriate, from where it is not clear if the evaluation can be trusted or interpreted, and thus the conclusions/contributions are not robust. Additionally I believe the paper's motivations lie in faulty assumptions about thresholds' behavior, and this is not discussed or argued further, this is a severe weakness in the narrative, specially since the authors could have argued why this assumption makes sense and in which cases it holds.

Unfortunately I do not think a revision can fix this paper due to the number of issues, and I can only recommend rejection.

---

### Official Review · Reviewer_7v19 · 2022-10-25

**Confidence:** 3
**Correctness:** 2
**Technical Novelty And Significance:** 1
**Empirical Novelty And Significance:** 1
**Recommendation:** 1

**Clarity, Quality, Novelty And Reproducibility:**

- The clarity of the paper is poor overall (see Weakness #1 above). The explanation of dataset splits in the 3 cases would be much clearer when shown as a figure.
- The paper does not have much empirical novelty over existing OOD detection benchmarks such as OpenOOD.
- The authors have not included their code in the supplementary or mentioned its release in the text, so the reproducibility is unknown.

**Strength And Weaknesses:**

Strengths:
- OOD detection is an important problem in machine learning.

Weaknesses:
1. The presentation of the paper is extremely poor. There are several spelling and grammar mistakes in almost every section. Figures 2-4 are also poorly presented, and the authors should consider focusing on a smaller set of results and showing it as a table.

2. The authors do not sufficiently motivate why a new OOD detection benchmark is necessary. As such, it is unclear to me what the significance of the paper is. What is the issue with existing benchmarks which the authors discuss in Section 2.2? Though the authors spend much of their attention on the dataset split, it seems that Case 2 is unlikely in practice as models are typically trained on one dataset, and Case 3 is a less useful version of Case 1, as it hides the per-OOD-dataset performance.

3. The authors do not show confidence intervals for any of their results. As such, it is not possible to conclude whether one method significantly outperforms another.

4. The authors focus a lot of attention on the F1 score as a metric, which assumes a fixed threshold on the OOD score. However, I believe that F1 is less useful as a metric as the threshold is highly dependent on the specific use case -- the relative cost of a FN and FP are highly dependent on where and how the model is used.

5. The authors do make a few interesting findings, such as the fact that existing OOD benchmarks are too easy, or the disagreement between AUROC and AUPRC. However, they are not very well explored, and do not provide much insight. The paper would be better served with more rigorous justifications such as those in [1].

6. The set of methods which the authors consider is rather small, relative to other benchmarks such as OpenOOD [2].

7. It seems to me that the support of the distribution of images from Places and TinyImageNet are very similar, so I am not convinced that one dataset can be considered OOD relative to the other.

[1] https://arxiv.org/pdf/2107.06908.pdf

[2] https://arxiv.org/pdf/2210.07242.pdf

**Summary Of The Paper:**

The authors propose a new benchmarking suite for OOD detection. It consists of five different datasets, and OOD detection tasks are constructed by taking different splits of the datasets as ID and OOD. The authors evaluate the performance of five OOD detection methods on their benchmark across four metrics, finding that the performance of OOD detection methods is highly dependent on the ID/OOD dataset split.

**Summary Of The Review:**

I believe that the paper requires large improvements in many aspects before it would be ready for publication. Therefore, I recommend rejection.

---

### Official Review · Reviewer_6ZgR · 2022-11-04

**Confidence:** 2
**Correctness:** 1
**Technical Novelty And Significance:** 1
**Empirical Novelty And Significance:** 1
**Recommendation:** 3

**Clarity, Quality, Novelty And Reproducibility:**

Writing clarity and quality needs to be improved. It is unclear how much novelty this work has without a readable rewriting of the whole draft. Reproducibility is poor as no much experimental details are included in the current draft.

**Strength And Weaknesses:**

## Strength

Out of distribution detection is an important problem in machine learning. A comprehensive benchmark that allows meaningful comparison across different OOD algorithms is highly valuable to the community.

## Weaknesses

Writing is extremely poor. Grammar and language issues are everywhere, rendering this work basically unreadable.

**Summary Of The Paper:**

This work provides a new benchmark for out of distribution detection with deep neural networks. Authors choose a total of 5 datasets and split them in different ways to create in-distribution (ID) and out-of-distribution (OOD) datasets. A total of three evaluation criteria are considered for ranking different algorithms. Authors find that the benchmarking results are highly sensitive to different settings.

**Summary Of The Review:**

Unsatisfying writing quality makes this work hard to understand. Paper is focused on evaluating existing methods on existing datasets, with existing evaluation metrics. No clear motivation on why the benchmark proposed in the work is necessary or novel.

---

### Decision · Program_Chairs · 2023-01-20

**Decision:**

Reject

**Justification For Why Not Higher Score:**

N/A

**Justification For Why Not Lower Score:**

N/A

**Metareview: Summary, Strengths And Weaknesses:**

The purpose of this paper is to investigate OOD detection by evaluating existing algorithms on a variety of datasets. The reviewers and AC agree that the paper is not ready for submission